



**Free Tropospheric Aerosols at the Mt. Bachelor Observatory:**
**More Oxidized and Higher Sulfate Content Compared to**
**Boundary Layer Aerosols**
Shan Zhou[1], Sonya Collier[1], Daniel A. Jaffe[2,3], Qi Zhang[1*]
[1]Department of Environmental Toxicology, University of California, Davis, CA 95616, USA
[2]School of Science, Technology, Engineering, and Mathematics, University of Washington Bothell, Bothell, WA,
USA
[3]Department of Atmospheric Sciences, University of Washington, Seattle, WA, USA
*Correspondence to*: Qi Zhang (dkwzhang@ucdavis.edu)
**Abstract**
Understanding the properties and lifecycle processes of aerosol particles in regional air masses is crucial for
constraining the climate impacts of aerosols on a global scale. In this study, characteristics of aerosols in the boundary
layer (BL) and free troposphere (FT) of a remote continental region in the western US were studied using a high-
resolution time-of-flight aerosol mass spectrometer deployed at the Mount Bachelor Observatory (MBO; 2763 m
a.s.l.) in central Oregon in summer 2013. In the absence of wildfire influence, the average ($\pm$ 1$\sigma$) concentration of
non-refractory submicrometer particulate matter (NR-PM$_1$) at MBO was 2.8 ($\pm$ 2.8) $\mu$g m$^{-3}$ and 84% of the mass was
organic. The organic aerosol (OA) at MBO from these clean periods showed clear diurnal variations driven by the
boundary layer dynamics with significantly higher concentrations occurring during daytime, upslope conditions. NR-
PM$_1$ contained a higher mass fraction of sulfate and was frequently acidic when MBO resided in the FT. In addition,
OA in the FT was found to be highly oxidized (O/C ~ 1.17) with low volatility. In contrast, OA associated with BL
air masses had an average O/C of 0.67 and appeared to be semivolatile. The significant compositional and physical
differences observed between FT and BL aerosols may have important implications for understanding the climate
effects of regional background aerosols.
**1      Introduction**
Atmospheric aerosols can scatter and absorb incident sunlight, therefore altering the radiation budget of the earth
directly. Depending on their chemical composition and microphysical properties, aerosol particles can also act as
cloud condensation nuclei and or ice nuclei and affect climate indirectly by altering the lifetime and optical properties
of clouds. Understanding the properties and the lifecycle processes of atmospheric aerosols is important for reducing
the uncertainties in aerosol climate forcing (Boucher, 2013).
Aerosols and their precursor gases are mostly emitted in the planetary boundary layer (PBL) but can be
transported into the free troposphere (FT) through convection and frontal uplift. In the FT, aerosols are subjected to
less efficient dry deposition and can have longer lifetimes than those in lower altitudes, facilitating regional



recirculation or long distance transport (Jaffe et al., 2005a; Dunlea et al., 2009; Sun et al., 2009). Under certain
atmospheric conditions, aerosols in the FT can be entrained into the BL, affecting remote regions where local
emissions may be minimal (Schroder et al., 2002; Timonen et al., 2013; Wang et al., 2016). A quantitative
understanding of aerosol properties and processes in regional background air masses and in the FT is necessary for
improving chemical transport models and global climate simulations.

High-altitude mountaintop observatories are important platforms for studying aerosols in regional air masses,

especially in the FT without the added expense and difficulty of making airborne measurements. Various mountaintop
sites have been operated in the US and Europe to perform long-term measurements on aerosol optical properties,
number, and size distributions and trace gases in continental background air masses (Jaffe et al., 2005b; e.g., Van
Dingenen et al., 2005; Reidmiller et al., 2010; Fischer et al., 2011; Hallar et al., 2011; Rose et al., 2015; Bianchi et al.,
2016; Hallar et al., 2016; Tröstl et al., 2016; Zhang and Jaffe, 2017). Aerosol chemical composition has also been
studied from high elevation sites, through both filter collection followed by offline analysis (e.g., Takahama et al.,
2011; Hallar et al., 2013; Dzepina et al., 2015) and real-time measurements using online aerosol mass spectrometers
(e.g., Zhang et al., 2007; Cozic et al., 2008; Sun et al., 2009; Fröhlich et al., 2015; Rinaldi et al., 2015; Freney et al.,
2016). These measurements have provided valuable information on the chemical and physical properties of remote
aerosols in the FT and PBL as well as how they are influenced by various sources (e.g., biomass burning, dust, and
biogenic emissions) and atmospheric processes (e.g., new particle formation, long-range transport, and cloud
processing).

The Mt. Bachelor Observatory (MBO) is a high-altitude atmospheric research site that has been utilized for

studying atmospheric chemistry in the western U.S. (Weiss-Penzias et al., 2006; Timonen et al., 2013). The
observatory is located at 2763 m above sea level at the summit of Mt. Bachelor, a dormant volcano in the Deschutes
National Forest in central Oregon (43.98˚ N, 121.69˚ W). Due to its elevation, MBO is situated in the FT at night and
is under the influence of upslope flow from the PBL air during the daytime (McClure et al., 2016). The remote
characteristics of the site makes MBO an ideal location for studying transported plumes, such as biomass burning
plumes from regional and distant sources (Jaffe et al., 2005b; Timonen et al., 2014; Briggs et al., 2016; Laing et al.,
2016; Zhang and Jaffe, 2017; Zhang, 2018) and long-range transport of Asian pollution in the spring (Jaffe et al.,
2005a; Weiss-Penzias et al., 2006; Fischer et al., 2010; Ambrose et al., 2011).

Continuous measurements of trace gases (e.g., ozone, carbon monoxide, carbon dioxide, mercury, nitrogen

oxides) and aerosol optical properties have been made at MBO since 2004.  In summer 2013, a high-resolution time-
of-flight aerosol mass spectrometer (HR-AMS; Aerodyne Research, Inc.) was deployed at MBO as part of the US
Department of Energy sponsored Biomass Burning Observation Project (BBOP) (Collier et al., 2016; Zhou et al.,
2017). This was the first real-time, highly time-resolved aerosol chemical measurement study performed at this site.
During this study, MBO was frequently impacted by transported wildfire plumes (Collier et al., 2016; Zhou et al.,
2017) but there were two periods, July 25 – 30 and August 17 – 21, when the site was not influenced by wildfires and
the concentrations of air pollutants remained low. Here, we focus on these clean periods to examine the chemical and
physical properties of regional background aerosols and to investigate the differences of aerosol characteristics and
processes in the PBL and the FT over Western US.



## 2 Methods

The HR-AMS was deployed at MBO from July 25 to August 25, 2013, as part of the BBOP campaign. Ambient aerosols were drawn through a $PM_{2.5}$ cyclone inlet and dehumidified by a Nafion dryer to eliminate potential RH effects on collection efficiency (CE). Treated particles then alternated between a heated thermodenuder (TD) line and an ambient bypass line every 5 minutes before entering the HR-AMS. Aerosol scattering (TSI nephelometer; 1 μm size cut), aerosol absorption (Tricolor Absorption Photometer, Brechtel; 1 μm size cut), CO and $CO_2$ (Picarro Cavity Ring-Down Spectroscopy G2502), $O_3$ (Dasibi), $NO_x$ (Air Quality Design 2-channel chemiluminescence), $NO_y$ (chemiluminescence), and peroxyacetyl nitrate (PAN; custom gas chromatograph) were also measured. Water vapor mixing ratios were calculated from the measured temperature, relative humidity (Campbell Scientific HMP 45C) and pressure (Vaisala PTB101B) following Bolton (1980), and typically agreed to within ± 15% and ± 0.3 g kg$^{-1}$ (Ambrose et al., 2011). Additional details of the instrumentation and methodology can be found in previous publications (Briggs et al., 2016; Collier et al., 2016; Zhou et al., 2017).

HR-AMS data were analyzed using established data analysis software tool Squirrel (v1.53) and Pika (v1.12; http://cires1.colorado.edu/jimenez-group/ToFAMSResources/ToFSoftware). A composition-dependent CE was applied based on the algorithm by Middlebrook et al. (2012) to account for possible CE changes induced by changes in particle phase in the AMS. A time-dependent gas phase $CO_2^+$ subtraction (Collier and Zhang, 2013) was performed to improve the quantification of organic aerosol (OA), which is critical for low aerosol loading conditions (Setyan et al., 2012). Elemental analysis of high-resolution mass spectra (HRMS) utilized both the Aiken-Ambient (AA) method (Aiken et al., 2008) and the Improved-Ambient (IA) method (Canagaratna et al., 2015).

Positive Matrix Factorization (PMF) was executed using the PMF2 algorithm (Paatero and Tapper, 1994) in the PET v2.05 program (Ulbrich et al., 2009) on the combined spectral matrices of organic and inorganic species (Sun et al., 2012; Zhou et al., 2017) during the clean periods without wildfire impact (i.e., July 25 – 30 and August 17 – 21). Organic ions at $m/z$ 12 – 180 and major inorganic ions, i.e., $SO^+$, $SO_2^+$, $HSO_2^+$, $SO_3^+$, $HSO_3^+$, and $H_2SO_4^+$ for sulfate, $NO^+$ and $NO_2^+$ for nitrate, $NH^+$, $NH_2^+$, and $NH_3^+$ for ammonium, and $HCl^+$ for chloride were included. The error matrix was pre-treated based on the procedures described in Ulbrich et al. (2009). After PMF analysis, the mass concentration of each OA factor was derived from the sum of organic signals in the corresponding mass spectrum after applying the default relative ionization efficiency (RIE = 1.4) for organics and the time-dependent CE. The solutions for 2 to 5 factors were explored with varying rotational parameters (-0.5 ≤ FPEAK ≤ 0.5, in increments of 0.1). Following the procedure listed in Table 1 in Zhang et al. (2011), PMF solutions were evaluated by investigating the key diagnostic plots, mass spectra, correlations with external tracers, and diurnal profiles. As shown in Fig. S1 in the supplementary material, the 2-factor solution showed relatively large residual while the 4-factor solution showed signs of factor splitting. The 3-factor solution resolved a less oxidized oxygenated OA (OOA) factor, a more oxidized OOA associated with some sulfate signals, and a sulfate-dominated OOA (Figs. S2 and S3). As the sulfate-dominated OOA accounted for only 3% of the total organic signal and its O/C and HRMS highly resembled those in the more-oxidized OOA factor (Fig. S3), these two factors were combined to form a so-called "highly oxidized OOA" factor which has an O/C of 1.17. Based on the chemical, physical characteristics and the volatility properties (see detailed discussions in Sect. 3.3), the less oxidized OOA was found to be semi-volatile OOA (SV-OOA) mainly associated with fresher



air masses from the BL whereas the highly oxidized OOA was comprised of low-volatility organic compounds (LV-
OOA) representing regional background OA in the FT. Furthermore, the time series and mass spectra of the SV-OOA
and LV-OOA derived here agreed well with the two background OOAs derived from PMF analysis of the whole
dataset, including periods influenced by wildfires (Zhou et al., 2017) (Figs. S4 and S5; $r^2 > 0.9$). All aerosol data in
this analysis are reported at ambient condition, except for aerosol light scattering, which is reported at STP (T = 273K
and P = 1013.25 hPa).
**3        Results and Discussion**
**3.1. Temporal and Diurnal Variations of Regional Background Aerosols Observed at MBO**

While observations at MBO were made continuously from July 25 to August 25, for this work, we use only data

from July 25 to 30 and August 17 to 21, 2013, which were relatively clear periods, free of wildfire influence. As
shown in Fig. 1, throughout the clear periods, the CO mixing ratio and submicron aerosol light scattering at 550 nm
($\sigma_{550nm}$) were below 120 ppb and 25 Mm$^{-1}$ at STP, respectively, similar to values previously observed at MBO under
clean conditions (Fischer et al., 2011; Timonen et al., 2014). The site was influenced by transported wildfire plumes
during the other periods of BBOP and air pollutant levels increased substantially, e.g., CO and $\sigma_{550nm}$ increased by up
to 8 –10 times compared to the clean periods and NR-PM$_1$ reached up to 140 µg m$^{-3}$ (Zhou et al., 2017). During the
clean periods, the HR-AMS indicator for biomass burning influence, namely the fraction of $C_2H_4O_2^+$ (*m/z* = 60.021)
signal over total OA ($f_{60}$), was below 0.3% (Fig. S6), confirming negligible influence from BB (Cubison et al., 2011).
Aerosol absorption data were available for the second clean period (August 17 – 21) and the average (± 1σ) EC mass
concentrations were estimated to be only 0.04 (± 0.14) µgC m$^{-3}$, further indicating a lack of BB influences.
Additionally, although winds at MBO showed a persistent westerly component (Fig. 1a and Fig. S7b), the bivariate
polar plot of NR-PM$_1$ concentrations exhibited a dispersed profile (Fig. S7c), indicating regional sources of aerosols
during the clean periods.

The average (± 1σ) concentration of NR-PM$_1$ (= sulfate + ammonium + nitrate + organics + chloride) during

these two clean periods was 2.8 (± 2.8) µg m$^{-3}$. OA was the largest PM$_1$ component, contributing on average ~ 84%
to the total NR-PM$_1$ mass, followed by sulfate (11%), ammonium (2.8%), nitrate (0.9%), and chloride (0.1%) (Fig.
S7a). Aerosol concentration and composition varied noticeably and showed diurnal changes that appeared to be mainly
driven by BL dynamics. This is because MBO sits in the FT at night but is influenced by air masses transported from
the PBL as the mixed layer height grows during the day. Indeed, the diurnal profile of the mixing-layer height retrieved
from the HYbrid Single Particle Lagrangian Integrated Trajectory (HYSPLIT) model (Draxler, 1998) shows that the
MBO is within the PBL between 12 – 8 pm PST (Fig. 2). In addition, previous studies at MBO have shown that water
vapor mixing ratio (H$_2$O$_{(g)}$) can be used to differentiate BL-influenced and FT air masses as FT conditions tend to be
very dry (Weiss-Penzias et al., 2006; Reidmiller et al., 2010; McClure et al., 2016; Zhang and Jaffe, 2017). H$_2$O$_{(g)}$ at
MBO varied from as low as 0.42 g kg$^{-1}$ at night to as high as 6.9 g kg$^{-1}$ during the day (Fig. 1b) and showed a strong
diurnal cycle similar to BLH and NO$_y$/CO (Fig. 2), another parameter for differentiating BL-influenced and FT air
(Stohl et al., 2002).





NR-PM$_1$ and $\sigma_{550nm}$ generally followed the temporal trend of H$_2$O$_{(g)}$ (Fig. 1b and c) and presented a pronounced
diurnal profile with substantial daytime enhancements (Fig. 2). The median mass concentration of NR-PM$_1$ was 0.5
µg m$^{-3}$ at night and increased by more than 10 times to 5.6 µg m$^{-3}$ in the afternoon. Similar temporal variations and
substantial daytime increases were observed for OA, nitrate, and gaseous pollutants such as CO, NO$_y$, and
peroxyacetyl nitrate (PAN) (Figs. 1 and 2), indicating that these species are primarily emitted or formed within the
BL and their concentrations at MBO are strongly influenced by BL dynamics. At night, the site is situated in the FT,
above the shallow nocturnal BL formed over the surrounding lower areas and disconnected from aerosol and gas
sources at the low altitudes. As the BL grows during the day, convective transport and thermal winds entrain pollution
from lower altitudes and increase air pollutants at the site. In contrast, sulfate and ammonium exhibited relatively
constant concentrations (Fig. 1e) and less pronounced diurnal patterns (Fig. 2). The weaker influence from BL
evolution indicates similar sulfate and ammonium concentrations in the BL and in the FT in the remote continental
region of the western US and is consistent with the relatively long atmospheric lifetime and the regional characteristics
of sulfate particles. O$_3$ and NO$_2$ mixing ratios also showed flat diurnal patterns (Fig. 2). However, a previous study at
MBO indicates that O$_3$ is typically higher in FT air masses (Zhang and Jaffe, 2017) but this depends on the air mass
origin and photochemical processing in both the BL and FT.
NR-PM$_1$ composition varied diurnally with a predominant organic composition during the day (up to 94% of
NR-PM$_1$ mass; Figs. 1 and 2). However, at night when the site was situated in the FT, sulfate was a major component
of aerosol (max = 83% of NR-PM$_1$; median = 37.6%; mean = 33%). OA during the clean periods at MBO was oxidized
with an average (± 1σ) O/C of 0.85 (± 0.36) and OM/OC of 2.26 (± 0.46). The degree of oxidation was in agreement
with regional background OA observed at other mountain sites such as Whistler Mountain in western Canada (Sun et
al., 2009), Rocky Mountains in Colorado US (Schurman et al., 2015), and Mt. Cimone in Italy (Rinaldi et al., 2015).
In addition, OA observed under the FT condition was overall more oxidized than those in the BL-influenced air
masses. For example, O/C peaked at night with a maximum value of 1.5 and reached a minimum of 0.7 in the afternoon
(Fig. 2). H/C anti-correlated with O/C with a reversed diurnal trend that peaked during daytime. As a result, the average
oxidation state of carbon (OS$_C$; = 2 O/C – H/C; (Kroll et al., 2011)) of OA at MBO during clean periods differs by 2
units between day and night (Fig. 2). These trends highlight the different chemical properties as well as atmospheric
ages of aerosols in the BL and the FT in this remote continental region in the western US. More discussions on the
differences between aerosols in BL and FT air masses are given in Section 3.4.
**3.2. Organonitrates and Organosulfates in Regional Background Aerosols**
Particulate organonitrates have been shown to make a significant contribution to submicron aerosol mass,
especially in rural and remote environments during summertime (Setyan et al., 2012; Fry et al., 2013; Kiendler-Scharr
et al., 2016; Zhou et al., 2016). In this study, organonitrates were observed and appeared to account for most of the
NO$^+$ and NO$_2^+$ (major ions of inorganic and organic nitrates in HR-AMS) signals detected in NR-PM$_1$ during the clean
periods. This is because the signal ratios of NO$^+$ and NO$_2^+$ measured for MBO aerosols, which ranged between 2.0
and 34.4 (average = 7.5; Fig. S9a), are substantially higher compared to the ratio for pure ammonium nitrate particles
($R_{AN}$ = 1.78 ± 0.07). Previous studies reported that the NO$^+$/NO$_2^+$ ratio for organonitrates ($R_{ON}$) are ~2.25 - 3.7 times





higher than $R_{AN}$ (Fry et al., 2009; Farmer et al., 2010; Fry et al., 2013). Based on this information and using the
equation (1) reported in Farmer et al. (2010), we estimated that nearly all the $NO^+$ and $NO_2^+$ signals measured during
the clean periods were contributed by organonitrates ($RONO_2$) from the fragmentation of the nitrate functional group
($-ONO_2$). Assuming that organonitrate molecules on average contain one $-ONO_2$ functional group per molecule and
have an average molecular weight of 230 g mol$^{-1}$ (Lee et al., 2006; Fry et al., 2009), we estimated that the average
concentration of organonitrates was 0.13 ($\pm$ 0.12) µg m$^{-3}$ (Fig. S9b) and accounted for ~5% of the total OA mass at
MBO during the clean periods. Since MBO is situated in a forested region covered by coniferous trees at lower
elevations, the reactions of monoterpenes with nitrate radicals were likely an important source of the observed
particulate organonitrates (Fry et al., 2009; Fry et al., 2013; Boyd et al., 2015; Ng et al., 2017) in upslope daytime air.
The presence of organosulfur compounds is also confirmed based on the unambiguous detection of sulfur-
containing organic ions ($C_xH_yS_qO_z^+$) such as $CH_2SO_2^+$, $CH_3SO_2^+$, $CH_4SO_3^+$, $CH_3S^+$, $C_3H_5SO_2^+$, and $C_4H_5SO_2^+$.
Previous studies have shown that $CH_2SO_2^+$, $CH_3SO_2^+$, and $CH_4SO_3^+$ are HR-AMS signature ions for methanesulfonic
acid (MSA) (Ge et al., 2012). In this study, the three ions correlate (r = 0.50 – 0.71; Fig. S10) with signal ratios close
to those observed for pure methanesulfonic acid (Ge et al., 2012). This is an indication that mesylate ($CH_3SO_3^-$, the
deprotonated anion of MSA) was present in the regional background aerosols in the western US. Based on the
fragmentation pattern of MSA, where $CH_3SO_2^+$ intensity contributed 8.7% to total MSA fragments in the HR-AMS
(Ge et al., 2012), we estimated that the average MSA mass concentration was 6.7 ($\pm$ 7.2) ng m$^{-3}$, making up ~ 0.3%
of the total OA mass during the clean periods.
Oceans are generally considered dominant sources of dimethyl sulfide (DMS) and therefore its oxidation product
MSA. However, the Pacific Ocean is 195 km to the west of MBO whereas the bivariate polar plot of MSA revealed
that high concentrations were associated with winds from the east and the south (Fig. S11). This is an indication that
the sources of MSA were mostly continental. In addition, MSA concentrations showed a clear diurnal cycle with a
substantial daytime increase (Fig. 2), which suggests significant sources from the PBL. Aerosols in the PBL over this
region likely have negligible oceanic influences since the Cascades mountain range lies between the Pacific Ocean
and Mt. Bachelor and may obstruct surface wind bringing marine emissions inland. DMS can be emitted from a wide
range of terrestrial sources including soil, vegetation, freshwater wetland, and paddy fields (Watts, 2000 and referencer
therein). Furthermore, the maximum MSA/SO$_4$ ratio was approximately 0.081, much lower than those observed in
marine aerosols (e.g., average = 0.23 in sub-Arctic North East Pacific Ocean; (Phinney et al., 2006)). The MSA/SO$_4$
ratios are much lower in terrestrial regions, e.g., 0.01 - 0.17 in Fresno where MSA was attributed to non-marine sources
(Ge et al., 2012; Young et al., 2016) and 0.007 - 0.15 along the Atlantic coast under continental influences (Zorn et
al., 2008; Huang et al., 2017).
**3.3. Sources and Processes of Aerosols in the Remote Region of the Western US**
PMF analysis was performed on the NR-PM$_1$ mass spectra acquired during the clean periods to further elucidate
the sources and processes of the regional background aerosols observed at MBO. Two OA factors were identified,
including an intermediately oxidized, semi-volatile OOA (SV-OOA, O/C = 0.67; H/C = 1.57) and a highly oxidized,
low volatility OOA (LV-OOA, O/C = 1.17 $\pm$ 0.08; H/C = 1.18 $\pm$ 0.03). No hydrocarbon-like (HOA) factor was





identified during the clean periods, which is consistent with a low abundance of $C_4H_9^+$ signal (0.13% of total OA
signal), a tracer ion for primary OA from vehicle emissions (Collier et al., 2015). In addition, $f_{60}$ was constantly lower
than 0.3% (Fig. S6 and Fig. S13b), indicating a lack of BB influence (Cubison et al., 2011). These results indicate the
absence of primary aerosol sources at MBO during clean periods.
SV-OOA, which on average accounted for 70% of total OA mass at MBO during clean periods (Fig. 3c), showed
temporal features that indicate a strong influence from BL dynamics. Particularly, SV-OOA correlated well with CO,
nitrate, and MSA (r = 0.7 – 0.84) and exhibited a pronounced diurnal cycle that increases between 9:00 – 10:00, peaks
around 15:30 (PST), and decreases to a very low concentration ($\sim$ 0.1 $\mu$g m$^{-3}$) at night (Fig. 2). The SV-OOA mass
spectrum displayed the characteristics of secondary OA (SOA) with two dominant oxygenated ions, $C_2H_3O^+$ (m/z =
43.018) and $CO_2^+$ (m/z = 43.989) (Fig. 3a). The signal intensity of $C_2H_3O^+$ is similar to that of $CO_2^+$ and the SV-OOA
spectrum comprises relatively abundant $C_xH_y^+$ and $C_xH_yO_1^+$ ions (Fig. 3a). These features, as well as an average O/C
of 0.67, indicate that SV-OOA was moderately oxidized and was likely not very aged.
The SV-OOA spectrum showed a significant $C_7H_7^+$ signal (m/z = 91.055), which was proposed as an indicator
for the presence of β-pinene + $NO_3$ reaction products (Boyd et al., 2015). Elevated $C_7H_7^+$ is an AMS spectral feature
for biogenic SOA observed both in ambient air and in chambers experiments (Kiendler-Scharr et al., 2009; Sun et al.,
2009; Robinson et al., 2011; Setyan et al., 2012; Budisulistiorini et al., 2015; Chen et al., 2015). In fact, the SV-OOA
spectrum of this study is highly similar to the spectra of biogenic SOA observed previously in both laboratory and
field studies. These findings as well as the fact that organonitrates were predominantly associated with SV-OOA (Fig.
S12) indicate that the SV-OOA observed in this study likely represented biogenic SOA formed at lower altitudes in
the region and transported upward to the site by thermal winds during the day.
LV-OOA, which accounted for an average 30% of total OA mass, likely represented more aged SOA in the
regional background air. It exhibited a much less pronounced diurnal trend than SV-OOA (Fig. 2) and presented as a
major OA component during most nights when the site was in the FT (Fig. 1g). These results suggest that LV-OOA
likely represents OA in the FT, which can be long-distance transported and/or regionally recirculated due to longer
aerosol lifetime and higher wind speed in the FT. LV-OOA was highly oxidized with an average O/C of 1.17 (Fig.
3b) and contributed major fractions of highly oxygenated organic ions, e.g., $C_4H_3O_3^+$, $C_3H_5O_3^+$, and $C_6H_5O_3^+$, and
$CO_2^+$ and $CHO_2^+$ – HR-AMS signature ions for carboxylic acids (Fig. 3e). In contrast, nearly all the $C_8H_{11}^+$, $C_6H_{11}O^+$,
$C_5H_9^+$ and $C_3H_7^+$ signals were attributed to SV-OOA, so were a majority of the $C_4H_7^+$ (91%) and $C_2H_3O^+$ (86%) signals
(Fig. 3d and 3e). In addition, LV-OOA was tightly associated with sulfate (Fig. 3b and 3e), a secondary aerosol species
representative of aged, regional air masses. Furthermore, LV-OOA situates near the apex of the triangle region for
ambient OAs in the $f_{44}$ vs $f_{43}$ space (Fig. S13a), overlapping with the highly oxidized LV-OOA observed in various
environments (Ng et al., 2010) as well as highly aged OOAs observed at high altitude (Sun et al., 2009; Fröhlich et
al., 2015). These results together suggest that LV-OOA likely represented free tropospheric SOA in the western U.S
and were composed of highly oxidized organic compounds.



**3.4 Differences between Aerosols in BL and FT Air Masses**

Because MBO tends to sample the free troposphere air at night and the regional boundary layer air during the day, the measurement data are used to examine the differences between aerosols in both parts of the atmosphere during clean periods. Extensive work has been done to differentiate free tropospheric air from boundary layer-influenced air at MBO using water vapor chairlift soundings (Reidmiller et al., 2010) and other approaches (Weiss-Penzias et al., 2006; Fischer et al., 2010; Ambrose et al., 2011; McClure et al., 2016; Zhang and Jaffe, 2017). Zhang and Jaffe (2017) contributed to a more accurate monthly FT/BL-influence isolation at MBO based on comparison of MBO $H_2O_{(g)}$ distributions to the $H_2O_{(g)}$ soundings from Medford and Salem, Oregon, at equivalent pressure level. Based on this work we classify periods with $H_2O_{(g)}$ above the minimum monthly water vapor criteria value, 2.5 g kg$^{-1}$, and CO > 80 ppb as those dominated by "BL-influenced air" and the rest by "FT air". A discussion on the comparison between this method and using the estimated BL height as the differentiating criteria can be found in Section 1 of the Supplement.

The average concentration of NR-PM$_1$ under BL influences was 3.16 µg m$^{-3}$, approximately 4 times of that in the FT (0.85 µg m$^{-3}$). While OA mass was on average 6 times higher in the BL than in the FT (2.7 vs 0.34 µg m$^{-3}$), sulfate mass concentrations in these two types of air masses were similar (0.35 vs 0.33 µg m$^{-3}$). The stoichiometric neutralization of the inorganic components of NR-PM$_1$ was examined by comparing the molar equivalent ratio of ammonium ($[NH_4^+]$/18) and sulfate ($[SO_4^{2-}]$/48) since inorganic nitrate and chloride concentrations appeared to be negligible during clean periods. The ammonium-to-sulfate equivalent ratios in aerosols during clean periods varied between 0.005 – 1 (Fig. 4a), indicating that remote aerosols in the western US were frequently acidic in the summer. Most significant is that a substantial amount of FT aerosols (~ 78% of FT NR-PM$_1$ mass vs 16% of BL NR-PM$_1$ mass) exhibited an ammonium-to-sulfate equivalent ratio lower than 0.3 (Fig. 4a), indicating the presence of very acidic particles in the free troposphere. Acidic FT particles were also observed at various high-altitude regional background sites, such as Jungfraujoch (Cozic et al., 2008; Fröhlich et al., 2015), Puy de Dome station (Freney et al., 2016), Whistler mountain (Sun et al., 2009), and Mauna Loa (Hawaii, US) (Johnson and Kumar, 1991), and during airborne measurements in the upper troposphere of the tropics (Froyd et al., 2009) and the Arctic (Brock et al., 2011; Fisher et al., 2011).

MSA correlated with HR-AMS sulfate for different aerosol regimes with different slopes. As shown in Fig. 4b, BL-influenced aerosols showed a range of MSA/SO$_4$ ratios generally higher than FT aerosols, possibly owing to relatively abundant sulfate particles in the FT. When a polluted air mass is lifted out of the boundary layer, the already formed aerosol can be washed out due to precipitation during lifting while the less-soluble gas phase compounds such as SO$_2$ are not entirely removed. The resulting gas phase mixture is then relatively enhanced in SO$_2$. Consequently, during the subsequent regional transport in the FT, sulfate forms in larger concentrations than MSA.

In addition to aerosol chemical properties, the physical properties of MBO aerosols were examined. The average mass-based size distribution of NR-PM$_1$ during the clean periods displayed a broad feature extending from 100 to 1000 nm in vacuum aerodynamic diameter (D$_{va}$, Fig. 5). Aerosol composition varied as a function of size with larger particles (>200 nm) having a relatively larger sulfate contribution (12%) compared to smaller particles (<5%). Org43, the organic signal at $m/z$ = 43 (90% of which was $C_2H_3O^+$), presented a broad distribution peaking between 250 and 350 nm in D$_{va}$ (Fig. 5b). In contrast, Org44, the organic signal at $m/z$ = 44 (95% of which was $CO_2^+$), and sulfate





displayed distinctly narrower distributions peaking at a larger droplet accumulation mode close to 500 nm (Fig. 5b
and 5c). The similar size distribution of Org44 and sulfate and the tight correlation between their concentrations ($r^2 =$
0.61; Fig. S14) suggest that highly oxidized organics and sulfate had similar sources and processes and are possibly
internally mixed. In particular, the prominent droplet mode at 500 nm indicates an important influence of aqueous-
phase reactions on the production of sulfate and highly oxidized organics. Indeed, previous studies have shown that
aqueous-phase processing (i.e., fog and cloud droplets and aerosol phase water) have led to production of more
oxidized organics (Lee et al., 2011; Lee et al., 2012; Ervens et al., 2013)  in the droplet mode (Ge et al., 2012) and
that aqueous-phase production of sulfate is an important process in the atmosphere (e.g., Ervens et al., 2011). In
addition, a similar sulfate size distribution was observed at the peak of Whistler Mountain, which had frequent cloud
cover (Sun et al., 2009).

A distinctly different size distribution was observed for sulfate-containing particles in the FT, as shown in Fig

5c, where it exhibited a prominent mode at ~ 250 nm. One possible explanation is condensational growth of newly
nucleated particles in the FT. Scavenging can significantly remove larger particles, resulting in low particle surface
area that facilitates new particle formation (NPF) in the FT. Indeed, in-situ NPF events have been frequently observed
in the FT at locations such as the Storm Peak Laboratory (Hallar et al., 2011; Hallar et al., 2013) and Jungfraujoch
(Bianchi et al., 2016; Tröstl et al., 2016), and in clean areas such as Arctic (Tunved et al., 2013; Freud et al., 2017).
The formation and growth of new particles has also been observed over a broad region in the FT (Tröstl et al., 2016).
At MBO, daily thermal winds may have uplifted gas precursors from the BL to the FT, which can be further oxidized
in the FT and then condensed onto new particles, leading to the condensational growth of particles (Bianchi et al.,
2016). These observations shed light on the different sources and processes of aerosols in the BL and FT and suggest
that sulfate and organic aerosols were likely present in variable mixtures (i.e., both internal and external mixtures) at
MBO.

**3.5. Comparisons with Aerosols Observed at Other High-altitude Locations**

Figure 6 summarizes the average composition of NR-PM$_1$ measured using AMS or Aerosol Chemical Speciation

Monitors (ACSM) at various elevated regional background ground sites (Zhang et al., 2007; Sun et al., 2009; Freney
et al., 2011; Worton et al., 2011; Fröhlich et al., 2015; Rinaldi et al., 2015; Schurman et al., 2015; Freney et al., 2016;
Zhu et al., 2016; Xu et al., 2018) and by aircraft (Bahreini et al., 2003; Dunlea et al., 2009). All of these measurements
were conducted under conditions absent of biomass burning influence and representative of regional background
aerosols in the northern hemisphere. Mountain-top studies separated FT air based on BLH calculated from LIDAR
measurements (Freney et al., 2016) or tracers such as $^{222}$Rn concentrations and NO$_y$/CO and back trajectory analysis
(Fröhlich et al., 2015). The average NR-PM$_1$ mass concentrations across all sites was 5.1 (± 6.9) µg m$^{-3}$ and the value
in North America was 2.6 (± 1.6) µg m$^{-3}$. NR-PM$_1$ concentrations were, on average, substantially lower in the FT than
in the BL influenced air (0.68 ± 0.18 µg m$^{-3}$ v.s. 5.8 ± 7.3 µg m$^{-3}$), reflecting generally clean conditions in the FT.

A major fraction (27 – 84%; average = 51%) of the NR-PM$_1$ mass was organic matter at these remote high-

altitude locations (Fig. 6a). For the same site, marked chemical difference can be seen between aerosols in the FT and
the BL. At all sites,  FT aerosols contained a substantially higher mass fraction of sulfate (39 – 44%) compared to the



mixed BL/FT aerosols (11 – 35%). Aircraft measurements also showed consistent results of higher sulfate content in
aerosols at higher altitudes. For example, Bahreini et al. (2003) reported that the sulfate contribution to total NR-PM$_1$
over east Asia increased from 17.4% in the lower atmosphere (1-3 km) to 28.8% in layers > 3 km. In the FT over the
northeast Pacific, more than half of the background submicron mass was attributed to sulfate (Dunlea et al., 2009;
Roberts et al., 2010). Elevated sulfate layers were also clearly observed in the higher altitudes above Mexico City
(DeCarlo et al., 2008). As a result, the mass ratio of submicron sulfate to organics (SO$_4$/Org) showed significantly
higher values (0.72 to 1.1) in the FT air masses than those in the mixed layers (0.13 – 0.7; Fig. 6b)

The extent to which sulfate particles are neutralized has major implications for aerosol radiative forcing. The

average relative humidity at MBO was 25.6 (±8.9) % during the clean periods. Acidic sulfate aerosols are more
hygroscopic than ammonium sulfate (Taylor et al., 2017), thus uptake of gaseous ammonia by acidic particles may
produce solid ammonium sulfate at low relative humidity. The resulting decrease in aerosol water content both reduces
the direct radiative forcing of sulfate (Adams et al., 2001; Jacobson, 2001) and inhibits homogenous ice nucleation by
liquid sulfate-containing particles (Koop et al., 2000). In addition, solid ammonium sulfate aerosols can also be
effective heterogeneous ice nuclei for cirrus cloud formation (Abbatt et al., 2006). While mineral dust particles coated
with ammonium sulfate are efficient ice nuclei, those coated with sulfuric acid can lose their ice nucleating ability
(Eastwood et al., 2009).
**4. Summary and Conclusions**

Based on field observations at a remote high-altitude atmospheric research station - the Mt. Bachelor Observatory

(MBO, 43.98° N, 121.69° W, 2763 m a.s.l.) in central Oregon - we have characterized the chemical and physical
properties of aerosols in the boundary layer and free troposphere air under clean conditions in the absence of wildfire
influences in the western US. Water vapor mixing ratio, a tracer used to segregate FT and BL-influenced air masses
at MBO, showed a strong diurnal cycle. Dry free tropospheric conditions were frequently observed at night, whereas
humid boundary layer influenced air was often observed during the day. The average (± 1σ) NR-PM$_1$ mass
concentration during the entire clean period was 2.8 (± 2.8) µg m$^{-3}$, with OA dominating the NR-PM$_1$ composition (~
84%) followed by sulfate (11%). OA, nitrate, and MSA displayed clear diurnal cycles with substantial daytime
increases, suggesting significantly higher mass concentrations in the BL than in the FT.

Strong diurnal patterns driven by the boundary layer dynamics were also observed in aerosol chemical

composition. NR-PM$_1$ contained a significantly higher mass fraction of sulfate (up to 83% of NR-PM$_1$ mass) and was
frequently acidic at night when MBO resided in the FT. In addition, nighttime free tropospheric OA was found to be
more oxidized. PMF analysis identified two types of OOA representing the regional background OA in the western
US: a LV-OOA (30% of OA mass) that was highly oxidized (O/C = 1.17) and comprised of low-volatility organics,
representative of SOA in the free troposphere; and an SV-OOA (70% of OA mass) that was intermediately oxidized
(O/C = 0.67) and appeared to be semivolatile, representative of biogenic SOA in the BL. NR-PM$_1$ observed at other
high-altitude locations in the world under regional background conditions were summarized. These results highlight
the significant compositional differences between FT and BL aerosols in that the FT aerosols are significantly more



oxidized and contain a higher fraction of sulfate. These observations may have important implications for
understanding the climate effects of aerosols in remote regions.
**Data availability**
Data presented in this manuscript are available upon request to the corresponding author.
**Acknowledgements**
This work is supported by US Department of Energy Atmospheric System Research Program, Grant No. DE-
SC0014620. Shan Zhou acknowledges funding from the Chinese Scholarship Council (CSC) and the Donald G.
Crosby Fellowship and the Fumio Matsumura Memorial Fellowship from the University of California at Davis. The
Mt. Bachelor Observatory is supported by the National Science Foundation (grant #AGS-1447832) and the National
Oceanic and Atmospheric Administration (contract #RA-133R-16-SE-0758).

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



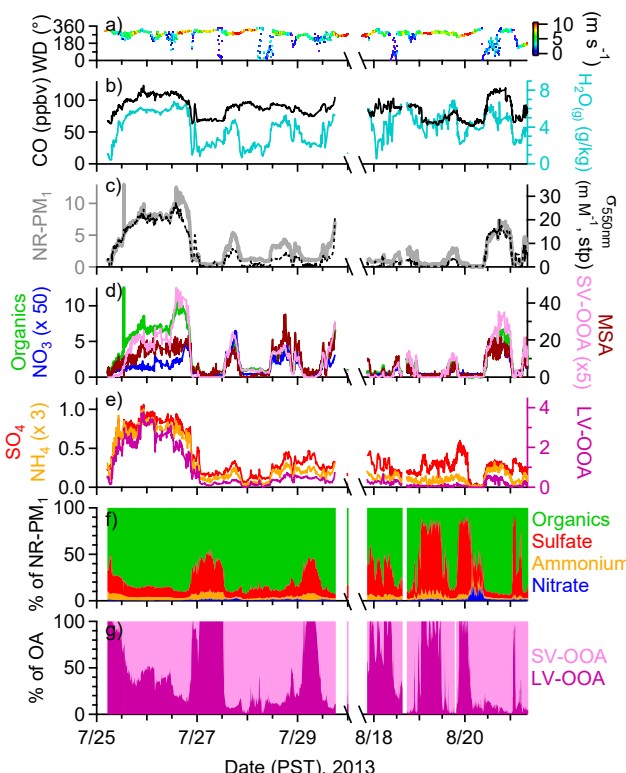


**Fig. 1**. Observations during two clean periods in summer 2013. Time series of (a) wind direction (WD) colored
by wind speed (WS), (b) mixing ratios of CO and water vapor ($H_2O_{(g)}$) (c - e) mass concentrations of NR-PM$_1$ speices
and OA factors (µg m$^{-3}$), and organic-equivalent mass concentration of MSA (ng m$^{-3}$) at ambient conditions,
submicorn aerosol light scattering at 550 nm ($\sigma_{550nm}$), (f) NR-PM$_1$ composition, and (g) OA composition.

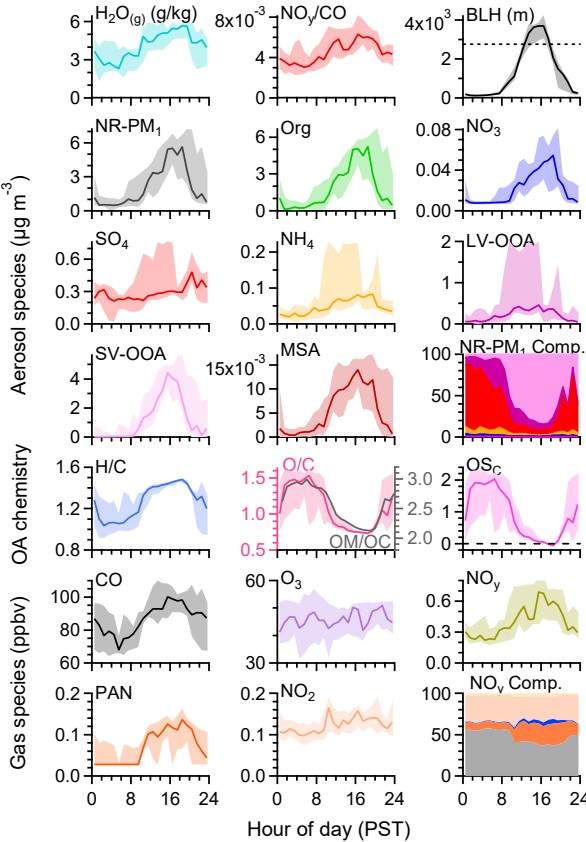


**Fig. 2**. Median diurnal cycles of water vapor ($H_2O_{(g)}$), $NO_y$/CO ratio (ppb/ppb), estimated boundary layer height (BLH), mass concentrations of NR-PM$_1$ species, OA chemistry, and mixing ratios of gas species during the clean periods shown in Fig.1 at MBO. MSA is in organic equivalent mass concentration. Oxidation state of carbon ($OS_C$) = 2 O/C – H/C. The shaded areas indicate the 75th and 25th percentiles. The diurnal cycle of NR-PM$_1$ composition displays the percent mass contributions, from top to bottom, of SV-OOA in light pink, LV-OOA in dark purple, sulfate in red, ammonium in orange, nitrate in blue, and chloride in purple. The diurnal cycle of $NO_y$ composition displays the percent mixing ratio contributions, from top to bottom, of NO in yellow, $NO_2$ in light orange, nitrate in blue, PAN in dark orange, and $NO_z$ (= $NO_y$ – NO – $NO_2$ – nitrate – PAN) in grey. Dashed line in BLH indicates the altitude of MBO (2763 m).





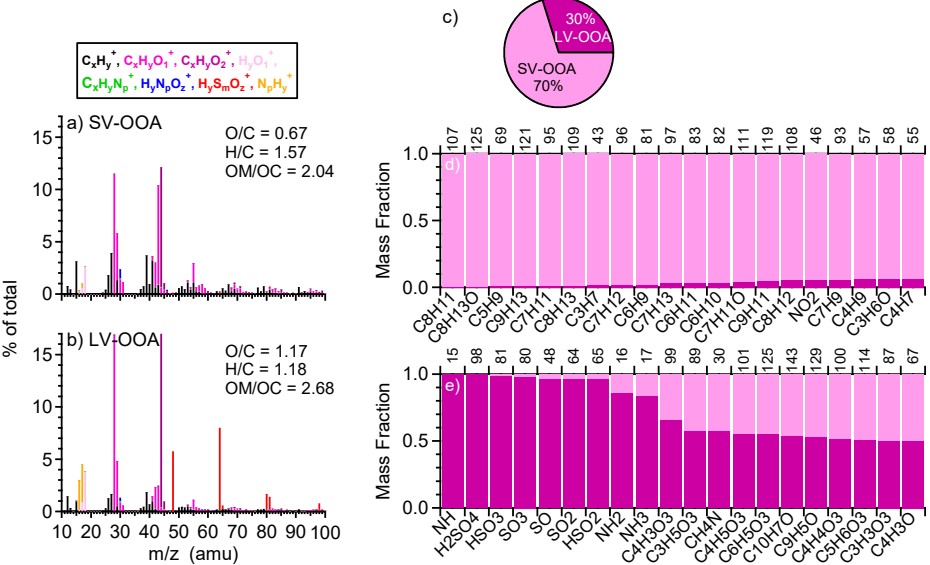


**Fig. 3.** High resolution mass spectrum of (a) SV-OOA and (b) LV-OOA colored by eight ion families. The
elemental ratios of OA determined using the IA method are shown in the legends. (c) Average OA composition. (d-e)
Ion signal distribution between SV-OOA and LV-OOA. Top 20 ions with greater fraction in SV-OOA in (d) and those
with greater fraction in LV-OOA in (e). The nominal masses of the ions are shown on the top axes of (d) and (e).




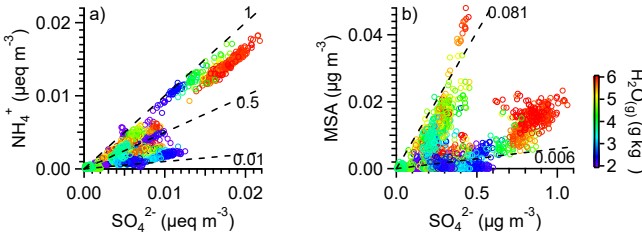


**Fig. 4**. (a) Ammonium molar equivalent concentration ($[NH_4^+]/18$) vs. sulfate molar equivalent concentration

($[SO_4^{2-}]/48$) and (b) MSA mass concentration vs. sulfate mass concentration colored by water vapor mixing ratio.

Dashed lines with different slopes are added for reference.



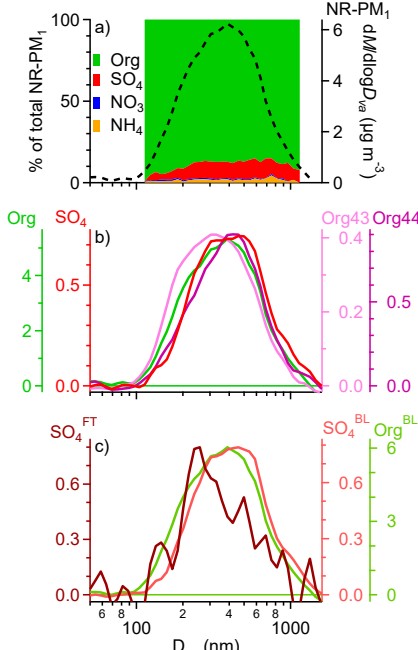


**Fig. 5**. (a) Size-resolved aerosol composition on the left axis, average size distributions of total NR-PM$_1$ mass

on the right. (b) Average mass size distributions of organics, sulfate, Org43, and Org44 during the clean periods. (c)

Average mass size distributions of FT sulfate and BL influenced sulfate and organics.



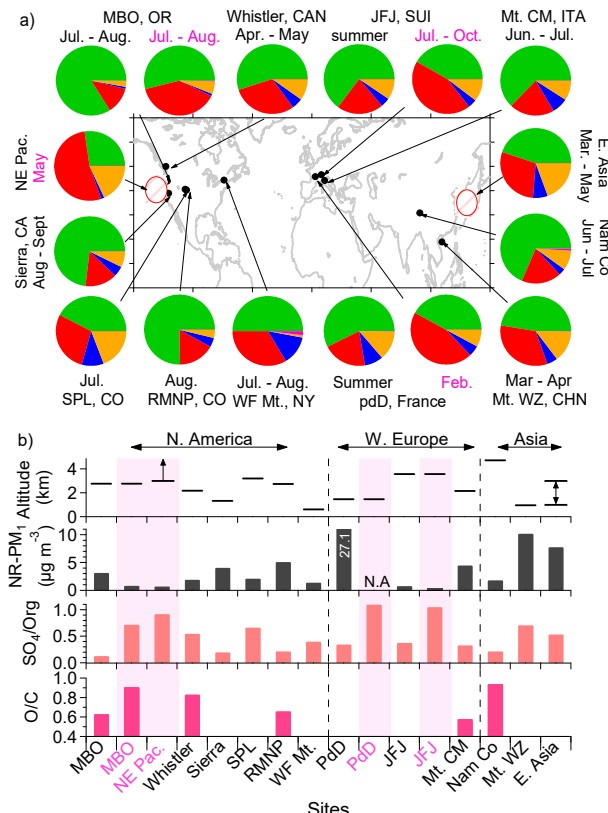


**Fig. 6.** (a) Location of selected high-altitude mountain sites and aircraft measurements of regional background aerosols in the world and the chemical composition of NR-PM$_1$ (details listed in Table S1 in the Supplement). Pie charts show the average chemical composition: organics (green), sulfate (red), nitrate (blue), ammonium (orange), and chloride (purple) of NR-PM$_1$. (b) Sampling altitude, average NR-PM$_1$ mass concentration, sulfate-to-organic mass ratio (SO$_4$/Org), and average O/C ratio of OA determined from the Ambient-Aiken method for each site. Colors of sampling period labels for pie charts in (a) and bottom axis labels for sites in (b) indicate air mass type: mixed BL/FT air (black) or FT air only (pink). Shaded pink bars in (b) indicate FT air.