# Peer review of "Free Tropospheric Aerosols at the Mt. Bachelor Observatory"

_Atmospheric Chemistry and Physics, 2018_

## Referee Comment (RC1) · Anonymous Referee #1 · 15 Nov 2018

This manuscript describes high resolution aerosol chemistry measurements made at the mount bachelor's observatory (MBO) during two periods in July and August of 2013. Aerosol chemistry and size distributions were measured using a high resolution aerosol mass spectrometer, and were operated alongside instruments characterizing gas-phase properties (NOY, CO, O3), and aerosol optical properties. As a result of its altitude and strong boundary layer dynamics the MBO site is often located in the free troposphere at night, providing the authors the opportunity to characterize FT composition. The objectives of this work were to describe the chemical and physical properties of FT aerosols and compare with those from the BL. This work is well prepared and described. The data set is a bit short and variable to generally describe FT aerosol

properties, but measurements at altitude sites are important to obtain a signature of FT aerosols.

General comments 1. How stable were the air mass sources during these two FT periods?

2. Page 5, Line 151: As discussed later on in the manuscript, this also suggest a supplementary source of sulfate aerosol particles in the FT, possibly from nucleation processes. Although no difference in the total mass of sulfate was observed during the day and night, were there differences in the signal ratios of the sulfate measured during the day (SO4_BL) vs night (SO4_FT).

3. The authors mention that the signal ratios of the sulfate peaks are similar to that of MSA. It is also mentioned that several sulfate peaks are associated with organic aerosols. Were there any trends in the PMF analysis that segregated the sulfate aerosol by diurnal patterns (the more oxidized OA vs the sulfate dominated OOA)? Were the signal ratios of the PMF sulfate factor similar to that of ammonium sulfate or similar to other types of sulfate.

4. From the PMF(ORG+INORG), were nitrate peaks associated with any of the organics, and was the PMF analysis capable of extracting an organic nitrate (ON) factor that could be compared to the ON extracted using the Farmer et al., method?

5. As shown in recent publications at both JFJ (Hermann et al., 2015) and at the PdD (Farah et al., 2018), the properties of FT aerosols can vary considerably depending on its last contact with the boundary layer. Similar analysis of air mass back trajectories on this data set would provide clear characteristics of FT aerosols at the MBO.

6. In the Hermann and Farah studies, the change in the size distribution for FT aerosols has been used as a tracer for FT air masses. In Figure 5, only average size distribution for the whole period are shown. Can the authors include the average size distributions for FT periods? Why are organic size distributions in the FT not included here? Can

they be magnified? or are the signal to noise ratios too low?

Herrmann, E.; Weingartner, E.; Henne, S.; Vuilleumier, L.; Bukowiecki, N.; Steinbacher, M.; Conen, F.;Collaud Coen, M.; Hammer, E.; Jurányi, Z.; et al. Analysis of long-term aerosol size distribution datanfrom Jungfraujoch with emphasis on free tropospheric conditions, cloud influence, and air mass transport. J. Geophys. Res. Atmos. 2015, 120, 9459–9480.

Farah, Antoine; Freney, Evelyn; Chauvigné, Aurélien; Baray, Jean-Luc; Rose, Clémence; Picard, David; Colomb, Aurélie; Hadad, Dani; Abboud, Maher; Farah, Wehbeh; Seasonal variation of aerosol size distribution data at the Puy de Dôme station with emphasis on the boundary layer/free troposphere segregation,Atmosphere, 2018, 9,7,244.

7. For the comparison with other "altitude" sites, the authors could also take into account measurements from:

Minguillón, M.C., Ripoll, A., Pérez, N., Prévôt, A.S.H., Canonaco, F., Querol, X., Alastuey, A., 2015. Chemical characterization of submicron regional background aerosols in the western Mediterranean using an Aerosol Chemical Speciation Monitor. Atmos. Chem. Phys 15, 6379–6391. doi:10.5194/acp-15-6379-2015

Ripoll, A., Minguillón, M.C., Pey, J., Jimenez, J.L., Day, D.A., Sosedova, Y., Canonaco, F., Prévôt, A.S.H., Querol, X., Alastuey, A., 2015. Long-term real-time chemical characterization of submicron aerosols at Montsec (southern Pyrenees, 1570 m a.s.l.). Atmos. Chem. Phys 15, 2935–2951. doi:10.5194/acp-15-2935-2015

8. Table S1, Figure 6: Why is there no NR-PM1 mass concentrations for pdD 2012?

9. Figure 6: O/C only 6/16 of the listed stations have values. Why so? It would be interesting to have this sort of overview of measurements? Can this information be obtained through making contact with the different stations?

10. Figure 6: It seems strange that the puy de Dome has such high concentrations

(higher than those in E.Asia (by a factor of 3!!)). Is this representative of the station or is it a particular period that was advected by unusual air masses.

Minor comments:

Page 2, Line 35: This statement is important and useful, but can the authors include some references based on recent simulations that show the need to improve our knowledge of regional and FT aerosol in order to improve chemistry transport models?

Page 6, Line 187, and Page 7, Line 239 (and elsewhere): Can you provide the m/z values for each of these fragments.

Page 7, Line 226: The m/z 91 can also be associated with fragments of primary anthropogenic OA. How can the authors be sure that this fragment is solely biogenic? It has previously been illustrated that the situation of the f44 and f43 on the triangle plot could indicate the source/type of the aerosol, with biogenic aerosols being situated on the right hand side of the triangle (Jimenez et al., 2009; Ng et al., 2010). Can this be illustrated and commented in Figure S13.

Figure 1: Can the authors identify the FT periods in Figure 1.

Figure 2: NR-PM1 Composition. What does the y-axis represent here, I presume fractional contribution. Here we see that, at night, in the FT, more than 90% of the particle composition is SO4 aerosols. In

Figure 1, we observe this to be the case in only 2 of the 9 nights. Is this figure an average of all data shown in Figure 1?

Figure S13: The main text refers to LV-OOA and SV-OOA, but in this figure the OOA are labeled BL-OOA and FT-OOA.

Jimenez, J.L., et al. Evolution of organic aerosols in the atmosphere. Science 326, 1525–9. doi:10.1126/science.1180353

Page 8, Line 264: Please include the ratios expected for acidic and for neutralized

conditions. Why not include the NH4 predicted to NH4 measured plots?

---

## Referee Comment (RC2) · Anonymous Referee #2 · 21 Nov 2018

The manuscript submitted by Shan Zhou and coauthors describes in detail a subset of a NR-PM1 aerosol chemistry dataset acquired at a high altitude sampling site, Mount Bachelor Observatory during the summer of 2013. It focuses on a few days where no apparent fire influence was observed at the site and uses this data to:

- Describe typical background summer conditions at MBO - Analyze how boundary layer dynamics modulate the chemical characteristics of the aerosol observed. - It then puts these measurements into a larger context of other NR-PM1 measurements at high altitude sites as well as aircraft observations.

The available field data presented is fairly sparse and this affects the robust-

ness/representativeness of the conclusions, as discussed below. Nevertheless, the authors provide adequate context in most cases to support their findings. The technical quality of the analysis is excellent, and provides additional insights into sources composition that are often missing in comparable publications. Given that these are the first AMS measurements under background conditions at a key high altitude site in North America, these results are important and useful, despite the small coverage.

The classification of air masses into BL/FT used in the manuscript is based on previous publications by the same authors and gives sensible results, although some more detail on the impact of the uncertainty of the assignment on the conclusions would be desirable. The summary of AMS observations at Mountain Sites is a useful addition and valuable for future high altitude studies. Also in this case the sample is at present fairly small, especially if one focuses on measurements that are clearly of free tropospheric air. May this publication help encourage further contributions.

Major comments:

Line 66: It would be useful to know what the criteria were to determine the clean/remote analysis periods. Line 118 mentions that <120 ppb CO and <25 Mm-1 at STP were observed, but those seem more descriptive (it clearly worked) than prescriptive. Can the authors explain if this based on back trajectories? Chemical markers? Other?

Line 110: Given that the two factors found are in good agreement with the factors found in Zhou et al, 2017, it would seem that focusing on the temporal evolution of these two for the full six week deployment would be a good way to improve statistics, especially regarding the BL/FT split (which should not really depend on the presence or absence of additional BB aerosol). PMF analysis in the presence of very large BB plumes can be challenging, but the analysis presented in Zhou et al, 2017 seems robust enough that this could work, and considerably strengthen the findings regarding e.g. daily trends and average FT composition. So I would encourage the authors to consider this possibility.

[Figure]

Line 151: The authors write that NH4 and SO4 are relatively comparable in the FT and BL, but this seems to contradict the acidity gradient described later (based on NH4/SO4 molar ratios), please clarify. Furthermore, the conclusion that the sources of SO4 are the same in the FT and BL is not really supported by the different size distributions observed (and the explanation given there). Given that sulfate concentrations in the BL are going to be a strong function of BL height and FT exchange, a direct comparison of these concentrations is not very meaningful, suggest removing.

Line 166/Fig 2: Constructing diurnal profiles based on six days of data is less than ideal, and the trends are somewhat obscured by the different times at which the PBL rose above the sampling site . Again, I suggest extending the PMF analysis for the full length of the deployment (Line 220) to spot check at least for some of the variables the robustness of the trends shown. That might also allow to filter for days where the BL/FT switch happened at consistent times and hence improve the robustness of the aerosol trends

Line 186: It should be mentioned that Lee et al, "Substantial secondary organic aerosol formation in a coniferous forest: observations of both day- and nighttime chemistry", Atmos. Chem. Phys., 16, 6721–6733, 2016, doi:10.5194/acp-16-6721-2016, reported the same trends in particulate organic nitrated at the Whistler "middle altitude site" with likely very similar forests sources as at MBO . The authors may also consider adding this dataset to Fig 6 as well, although the lower sampling altitude makes it less comparable to other high altitude observations.

Line 196: The MSA analysis is well done and a nice addition to the manuscript. Are the authors aware of the study by Sorooshian et al, "Surface and airborne measurements of organosulfur and methanesulfonate over the western United States and coastal areas" J. Geophys. Res. Atmos., 120(16), 8535–8548, doi:10.1002/2015JD023822, 2015? They measured MSA and OS at ground sites near MBO during the Summer of 2013 as well, and found broadly similar concentrations, consistent with MSA being made in the continental BL.

Section 3.4: The diurnal profiles show a fairly clear "transition zone" between FT and BL influence, and including/excluding those periods will likely strongly affect the averages found (see also Wagner et al, Atmos. Chem. Phys., 15, 7085–7102, 2015 as an example on the difficulty of properly defining the top of the BL). Such an analysis, either based on the diurnal profiles themselves or on the uncertainty of their threshold criteria, would help strengthen the confidence in the reported averages and trends, especially for the FT data.

Line 275-278: This seems mostly speculative. MSA could also be oxidized further to sulfate and/or partition to the gas phase instead of being washed out. Consider removing/shortening.

Line 295-306: Again this seems fairly speculative. There is no discernible enhancement of small particles in the (fairly noisy) sulfate size distribution presented, which is still within the envelope of the BL size distribution. So preferential activation/wash out of larger particles would suffice to account for the observed differences and nucleation is not needed to explain the difference. So consider shortening/removing.

Line 328-336: I am not following. If the average FT NH4/SO4 molar ratio is less than 0.3, the particles will be liquid bisulfate/sulfuric acid down to fairly low RH, hence the discussion of solid ammonium sulfate as presented does not seem really relevant to the findings. Please explain.

Minor comments:

Line 32: "At lower altitudes " instead of "in lower altitudes"

Line 39: While aircraft measurements are indeed expensive, I would argue that the main advantage of mountaintop observatories are long-term, continuous measurements that are invaluable for statistics. Obviously deploying complicated instrumentation such as the AMS can be a challenge for such sites as well, but I think it is worth mentioning this.

[Figure]

Line 40: Given the inclusion of the Whistler site, I would suggest replacing "US" with "North America"

Line 45: Suggest adding to your list of non-AMS particle measurements: L. Ahlm et al, "Temperature-dependent accumulation mode particle and cloud nuclei concentrations from biogenic sources during WACS 2010", Atmos. Chem. Phys., 13, 3393–3407, 2013, doi:10.5194/acp-13-3393-2013

Line 82: "the established data analysis tool" instead of "established data analysis tool"

Line 84: It would be useful to document the range of CEs observed. A histogram keyed by BL/FT influence would be appropriate, and would serve to highlight again the gradient in acidity between the different air masses sampled at MBO.

Line 131: Is the chloride associated with periods of somewhat larger inorganic nitrate? As Hu et al, Aerosol Sci. Technol., 51,735-754, doi:10.1080/02786826.2017.1296104 described, under such conditions chloride can be a vaporizer artifact, and is easily testable by reviewing the ammonium nitrate calibrations. Given the relative concentrations of chloride and nitrate, this is unlikely, but should be mentioned.

Fig 4: Suggest adding a timeseries of the ratios, with the FT periods clearly marked, that would make the point(s) better than these correlation plots.

Fig S13 (middle): Please add the background line from Cubison et al. It looks like your factors lie pretty much right on top of it, as in Zhou et al, 2017.

---

## Author Comment (AC1) · 21 Jan 2019

Response to reviewer comments on "Free Tropospheric Aerosols at the Mt. Bachelor Observatory: More Oxidized and Higher Sulfate Content Compared to Boundary Layer Aerosols"

by Shan Zhou et al.

We thank the reviewers for their thoughtful comments. We have carefully revised the manuscript accordingly. Our point-to-point responses can be found below, with reviewer comments repeated in black and author responses in blue. Changes made to the manuscript are in quotation marks.

**Author response to Anonymous Referee #1**

This manuscript describes high resolution aerosol chemistry measurements made at the mount bachelor's observatory (MBO) during two periods in July and August of 2013. Aerosol chemistry and size distributions were measured using a high resolution aerosol mass spectrometer, and were operated alongside instruments characterizing gas-phase properties (NOY, CO, O3), and aerosol optical properties. As a result of its altitude and strong boundary layer dynamics the MBO site is often located in the free troposphere at night, providing the authors the opportunity to characterize FT composition. The objectives of this work were to describe the chemical and physical properties of FT aerosols and compare with those from the BL. This work is well prepared and described. The data set is a bit short and variable to generally describe FT aerosol FT aerosols.

General comments
1. How stable were the air mass sources during these two FT periods?
The sources of air masses during these two FT periods were quite stable based on 3-day back-trajectories calculated for air masses arriving at MBO every hour. Specifically, the clean periods mostly showed trajectories from the west, originating from the Pacific Ocean.

2. Page 5, Line 151: As discussed later on in the manuscript, this also suggest a supplementary source of sulfate aerosol particles in the FT, possibly from nucleation processes. Although no difference in the total mass of sulfate was observed during the day and night, were there differences in the signal ratios of the sulfate measured during the day (SO4_BL) vs night (SO4_FT).
The signal ratios among sulfate peaks, i.e., mass spectral fragmentation pattern of sulfate, are consistent throughout the entire campaign, thus nearly identical between the FT and BL. Note that there were differences in sulfate/NR-PM$_1$ ratio between day and night. As shown in figure 2 and discussed in the last paragraph of section 3.1., sulfate was the dominant PM$_1$ component at night whereas organic aerosol dominated PM$_1$ composition during daytime. The mean SO4/NR-PM$_1$ ratio for FT periods (similar but not equal to nighttime) was 0.39, about 3.5 times of that for the BL periods (0.11).

3. The authors mention that the signal ratios of the sulfate peaks are similar to that of MSA. It is also mentioned that several sulfate peaks are associated with organic aerosols. Were there any trends in the PMF analysis that segregated the sulfate aerosol by diurnal patterns (the more oxidized OA vs the sulfate dominated OOA)? Were the signal ratios of the PMF sulfate factor similar to that of ammonium sulfate or similar to other types of sulfate.
The signal ratios of the sulfate peaks are actually different than those of MSA. In the PMF 3-factor solution, sulfate ions (i.e., SO$_x^+$) were associated with two factors, the more oxidized OA and sulfate-dominated OOA. The diurnal profiles of these two factors are not identical, but both are flat in comparison with that of SV-OOA, which showed a strong daytime enhancement.

The ratios among sulfate ions in the two OOA spectra are nearly identical and both are highly similar to that of ammonium sulfate.

4. From the PMF(ORG+INORG), were nitrate peaks associated with any of the organics, and was the PMF analysis capable of extracting an organic nitrate (ON) factor that could be compared to the ON extracted using the Farmer et al., method?

As shown in Fig. S12, 78% of the $NO^+$ and $NO_2^+$ signals were associated with SV-OOA and the rest 22% were associated with the LV-OOA factor. Together with the high $NO^+/NO_2^+$ ratios, this result indicates that the nitrate signals observed during clean periods were predominantly contributed by organic nitrates, which is consistent with the Farmer et al method.

We have revised text in the 3$^{rd}$ paragraph of Section 3.3, which now reads:
"These findings, together with the fact that organonitrates were predominantly associated with SV-OOA (e.g., 78% of the aerosol nitrate signal was attributed to SV-OOA; Fig. S12), indicate that the SV-OOA observed in this study likely represented biogenic SOA formed at lower altitudes in the region and transported upward to the site by thermal winds during the day. "

5. As shown in recent publications at both JFJ (Hermann et al., 2015) and at the PdD (Farah et al., 2018), the properties of FT aerosols can vary considerably depending on its last contact with the boundary layer. Similar analysis of air mass back trajectories on this data set would provide clear characteristics of FT aerosols at the MBO.

Hermann et al. (2015) and Farah et al. (2018) focused on characterizing aerosols in the free troposphere. Both studies used Lagrangian particle dispersion model to estimate the interaction of FT air mass with the BL and observed that particle number concentrations in the FT changed with time since the last contact with BL. While examining the influence of BL contact on FT aerosol characteristics at MBO would indeed be very interesting, using a Lagrangian particle dispersion model is beyond the scope of this work and analyzing the air mass back trajectories for this study may not work well since the trajectory model only allows for the estimation of time since last contact with BL every hour. In addition, the relatively short duration of this study period (~ 160 hours) may limit the statistical significance of the analysis results. Nevertheless, our approach of using water vapor mixing ratios to differentiate the FT and BL-influenced air masses is supported by extensive works previously conducted at MBO and serves well the goal of this present study, which focuses on characterizing chemical differences between aerosols in FT and in BL-influenced air masses. Details on the method are given in the first paragraph of section 3.4 and in Section 1 of the supplementary.

6. In the Hermann and Farah studies, the change in the size distribution for FT aerosols has been used as a tracer for FT air masses. In Figure 5, only average size distribution for the whole period are shown. Can the authors include the average size distributions for FT periods? Why are organic size distributions in the FT not included here? Can they be magnified? or are the signal to noise ratios too low?

We examined the average size distributions for both FT and BL-influenced periods. The size distributions of organic and sulfate in the BL-influence periods are shown in Fig. 5c. The size distribution of FT-sulfate is shown as well. However, the average organic size distribution during the FT periods is not shown because of low signal to noise ratio.

Herrmann, E.; Weingartner, E.; Henne, S.; Vuilleumier, L.; Bukowiecki, N.; Steinbacher, M.; Conen, F.;Collaud Coen, M.; Hammer, E.; Jurányi, Z.; et al. Analysis of long-term aerosol size distribution data from Jungfraujoch with emphasis on free tropospheric conditions, cloud influence, and air mass transport. J. Geophys. Res. Atmos. 2015, 120, 9459–9480.

Farah, Antoine; Freney, Evelyn; Chauvigné, Aurélien; Baray, Jean-Luc; Rose, Clémence; Picard, David; Colomb, Aurélie; Hadad, Dani; Abboud, Maher; Farah, Wehbeh; Seasonal variation of aerosol size distribution data at the Puy de Dôme station with emphasis on the boundary layer/free troposphere segregation,Atmosphere, 2018, 9,7,244.

7. For the comparison with other "altitude" sites, the authors could also take into account measurements from:
Minguillón, M.C., Ripoll, A., Pérez, N., Prévôt, A.S.H., Canonaco, F., Querol, X., Alastuey, A., 2015. Chemical characterization of submicron regional background aerosols in the western Mediterranean using an Aerosol Chemical Speciation Monitor. Atmos. Chem. Phys 15, 6379–6391. doi:10.5194/acp-15-6379-2015

Ripoll, A., Minguillón, M.C., Pey, J., Jimenez, J.L., Day, D.A., Sosedova, Y., Canonaco, F., Prévôt, A.S.H., Querol, X., Alastuey, A., 2015. Long-term real-time chemical characterization of submicron aerosols at Montsec (southern Pyrenees, 1570 m a.s.l.). Atmos. Chem. Phys 15, 2935–2951. doi:10.5194/acp-15-2935-2015

We thank the reviewer for recommending these references. The Ripoll et al. (2015) study, which was made at the summit of Montsec representing regional background conditions and covered mixed BL and FT periods in summer, has now been included in Fig. 6 and Table S1.  However, we decide not to include the Minguillon et al. (2015) study, which took place on the upper slopes at an elevation of 720 m a.s.l. downwind of Barcelona urban areas, since its results are considered less representative of regional background conditions compared to the other high-altitude sites shown in Fig. 6.

8. Table S1, Figure 6: Why is there no NR-PM1 mass concentrations for pdD 2012?
We have added the average NR-PM$_1$ mass from pdD 2012 in Table S1 and Figure 6.

9. Figure 6: O/C only 6/16 of the listed stations have values. Why so? It would be interesting to have this sort of overview of measurements? Can this information be obtained through making contact with the different stations?
The reason is that only 9 out of 16 of the listed studies used HR-AMS, out of which only 6 published the average O/C ratios. We contacted the authors/stations of the 3 studies to request average O/C ratios but received no response.

10. Figure 6: It seems strange that the puy de Dome has such high concentrations (higher than those in E.Asia (by a factor of 3!!)). Is this representative of the station or is it a particular period that was advected by unusual air masses.
We took at closer look at the puy de Dome study and found that the BL data during summer 2010 was probably not representative of the average summertime BL aerosols at Puy de Dome. The study only covered one week of sampling and 80% of the air masses arriving at the site were from the more polluted continental sector. We have replaced it with the data collected for the same site during the 2012 winter campaign, which appeared to be more representative of the wintertime BL aerosol at the site.  Figure 1, Table S1, and relevant texts throughout the manuscript have been updated.

Minor comments:
Page 2, Line 35: This statement is important and useful, but can the authors include some references based on recent simulations that show the need to improve our knowledge of regional and FT aerosol in order to improve chemistry transport models?

We deduced the importance of properly representing FT aerosol chemistry in models. This sentence has now been revised:

"A quantitative understanding of aerosol properties and processes in regional background air masses and in the FT would be useful for improving chemical transport models and global climate simulations."

Page 6, Line 187, and Page 7, Line 239 (and elsewhere): Can you provide the m/z values for each of these fragments.
The m/z values for ion fragments have been added throughout the manuscript.

Page 7, Line 226: The m/z 91 can also be associated with fragments of primary anthropogenic OA. How can the authors be sure that this fragment is solely biogenic? It has previously been illustrated that the situation of the f44 and f43 on the triangle plot could indicate the source/type of the aerosol, with biogenic aerosols being situated on the right hand side of the triangle (Jimenez et al., 2009; Ng et al., 2010). Can this be illustrated and commented in Figure S13.
It is true that m/z 91 can also be associated with fragments of primary anthropogenic OA and that the mass fraction of m/z 91 to total organic ($f_{91}$) is not a unique tracer for BSOA (Ng et al., 2011). However, in unpolluted biogenic-rich environment, $f_{91}$ is useful for evaluating formation pathways of BSOA (Lee et al., 2016). Anthropogenic emissions have a negligible influence on MBO during this study and this conclusion is supported by aerosol composition measurements.

We have revised the discussions about $f_{91}$ in the revised manuscript and the 3$^{rd}$ paragraph of Section 3.3. now reads:

"The SV-OOA spectrum showed a significant $C_7H_7^+$ signal at $m/z$ = 91.055 ($f_{C7H7+}$ = 0.65%) and a spectral pattern highly similar to biogenic SOA observed from a plant chamber (Kiendler-Scharr et al., 2009). $C_7H_7^+$ was proposed as an indicator for the presence of β-pinene + $NO_3$ reaction products (Boyd et al., 2015) and elevated $f_{C7H7+}$ was previously observed in the AMS spectra of biogenic SOA both in ambient air and in chamber experiments (Kiendler-Scharr et al., 2009; Sun et al., 2009; Robinson et al., 2011; Setyan et al., 2012; Budisulistiorini et al., 2015; Chen et al., 2015). In addition, as shown in Fig. S13a, the SV-OOA of this study situates along the right leg of the triangle defined by worldwide ambient OA in the $f_{44}$ vs $f_{43}$ space. It has been illustrated previously that the $f_{44}$ vs. $f_{43}$ triangle plot could be used to indicate the source/type of the aerosols and that biogenic OA usually situate on the right hand side of the triangle (Jimenez et al., 2009; Ng et al., 2010). These findings, together with the fact that organonitrates were predominantly associated with SV-OOA (e.g., 78% of the aerosol nitrate signal was attributed to SV-OOA; Fig. S12), indicate that the SV-OOA observed in this study likely represented biogenic SOA formed at lower altitudes in the region and transported upward to the site by thermal winds during the day."

Figure 1: Can the authors identify the FT periods in Figure 1.
A color indicator has been added on the top of Figure 1 to signify FT and BL-influenced periods. The caption of Figure 1 has been updated accordingly.

Figure 2: NR-PM1 Composition. What does the y-axis represent here, I presume fractional contribution. Here we see that, at night, in the FT, more than 90% of the particle composition is SO4 aerosols. In Figure 1, we observe this to be the case in only 2 of the 9 nights. Is this figure an average of all data shown in Figure 1?
The y-axis represents the fractional mass contribution, as noted in the figure caption. Figure 2 shows that at night, $SO_4$ aerosol contributed up to 70% of the $NR-PM_1$ composition. For time from 0 to 5 am, sulfate fraction was on average ~ 60% only. This is consistent with figure 1.

Figure S13: The main text refers to LV-OOA and SV-OOA, but in this figure the OOA are labeled BL-OOA and FT-OOA.

Thanks for spotting the inconsistency.  Labels in Figure S13 have been changed to LV-OOA and SV-OOA.

Jimenez, J.L., et al. Evolution of organic aerosols in the atmosphere. Science 326, 1525–9. doi:10.1126/science.1180353

We have cited this reference.

Page 8, Line 264: Please include the ratios expected for acidic and for neutralized conditions. Why not include the NH4 predicted to NH4 measured plots?

Nitrate and chloride concentrations were negligible in the study period and a majority of the nitrate signal was likely contributed by organic nitrates. Thus, only sulfate was considered in the calculation of the predicted NH4. In this case, Figure 4a serves in the same way as the scatterplot of NH4 measured vs NH4 predicted.

The following sentence has been added:

"An ammonium-to-sulfate equivalent ratio of 1 suggests neutral particles whereas a ratio significantly lower than 1 suggests acidic particles.

**Anonymous Referee #2**

The manuscript submitted by Shan Zhou and coauthors describes in detail a subset of a NR-PM1 aerosol chemistry dataset acquired at a high altitude sampling site, Mount Bachelor Observatory during the summer of 2013. It focuses on a few days where no apparent fire influence was observed at the site and uses this data to: Describe typical background summer conditions at MBO - Analyze how boundary layer dynamics modulate the chemical characteristics of the aerosol observed. It then puts these measurements into a larger context of other NR-PM1 measurements at high altitude sites as well as aircraft observations. The available field data presented is fairly sparse and this affects the robustness/representativeness of the conclusions, as discussed below. Nevertheless, the authors provide adequate context in most cases to support their findings. The technical quality of the analysis is excellent, and provides additional insights into sources composition that are often missing in comparable publications. Given that these are the first AMS measurements under background conditions at a key high altitude site in North America, these results are important and useful, despite the small coverage. The classification of air masses into BL/FT used in the manuscript is based on previous publications by the same authors and gives sensible results, although some more detail on the impact of the uncertainty of the assignment on the conclusions would be desirable. The summary of AMS observations at Mountain Sites is a useful addition and valuable for future high altitude studies. Also in this case the sample is at present fairly small, especially if one focuses on measurements that are clearly of free tropospheric air. May this publication help encourage further contributions.

Major comments:
Line 66: It would be useful to know what the criteria were to determine the clean/remote analysis periods. Line 118 mentions that <120 ppb CO and <25 Mm-1 at STP were observed, but those seem more descriptive (it clearly worked) than prescriptive. Can the authors explain if this based on back trajectories? Chemical markers? Other?

The clean periods in this study were determined based on the AMS indicator for biomass burning influences, $f_{60}$, the fraction of $C_2H_4O_2^+$ (m/z = 60.021) signal over total OA. Data with $f_{60}$ lower than 0.3% (Cubison et al., 2011) were classified as clean periods. Concentrations of CO and aerosol particles were very low during these periods. In addition, 3-day back trajectories suggest that the air masses arriving at MBO during the clean periods had very stable source, mostly from the west, originating from the Pacific Ocean.

To make these points clearer, we have revised the first paragraph of section 3.1., which now reads:
  "While observations at MBO were made continuously from July 25 to August 25, for this work, we use only data from July 25 to 30 and August 17 to 21, 2013, which were classified as periods free of wildfire influence. The HR-AMS indicator for biomass burning influence, namely the fraction of $C_2H_4O_2^+$ (m/z = 60.021) signal over total OA ($f_{60}$), was used for differentiating wildfire influences. Periods with $f_{60}$ below 0.3% (Fig. S6) likely received negligible influence from BB (Cubison et al., 2011), thus were classified as clean periods. As shown in Fig. 1, throughout the clean periods, the CO mixing ratio and submicron aerosol light scattering at 550 nm ($\sigma_{550nm}$) were below 120 ppb and 25 Mm$^{-1}$ at STP, respectively, similar to values previously observed at MBO under clean conditions (Fischer et al., 2011; Timonen et al., 2014). The site was influenced by transported wildfire plumes during the other periods of BBOP and air pollutant levels increased substantially, e.g., CO and $\sigma_{550nm}$ increased by up to 8 –10 times compared to the clean periods and NR-PM$_1$ reached up to 140 µg m$^{-3}$ (Zhou et al., 2017). Aerosol absorption data were available for the second clean period (August 17 – 21) and the average (± 1σ) EC mass concentrations were estimated to be only 0.04 (± 0.14) µgC m$^{-3}$, further indicating a lack of BB influences. Additionally, although winds at MBO showed a persistent westerly component (Fig. 1a and Fig. S7b), the bivariate polar plot of NR-PM$_1$

concentrations exhibited a dispersed profile (Fig. S7c), indicating regional sources of aerosols during the clean periods."

Line 110: Given that the two factors found are in good agreement with the factors found in Zhou et al, 2017, it would seem that focusing on the temporal evolution of these two for the full six week deployment would be a good way to improve statistics, especially regarding the BL/FT split (which should not really depend on the presence or absence of additional BB aerosol). PMF analysis in the presence of very large BB plumes can be challenging, but the analysis presented in Zhou et al, 2017 seems robust enough that this could work, and considerably strengthen the findings regarding e.g. daily trends and average FT composition. So I would encourage the authors to consider this possibility.

We differentiate FT and BL-influenced air masses based on measured water vapor mixing ratio and CO concentration and classify periods with H2O(g) < 2.5 g kg-1 and CO < 80 ppb as "FT air" and the rest as "BL-influenced air". Therefore, the BL/FT split only applies to clean periods. In addition, due to substantially higher aerosol signals during BB influenced periods, we had to exclude some low loading periods (Org < 1.5 ug m$^{-3}$) from the full-length PMF analysis in order to allow the model to convergence. Approximately half of the clean period data were included in the full-length PMF analysis as a result. Fig. S4 and Fig. S5 demonstrated that the temporal trends of the two regional background factors from both PMF analysis (clean periods only and full-length) correlated tightly. This suggest that results derived from clean periods are statistically significant and robust.

We have clarified this point by revising this section, which now reads:
"Furthermore, the time series and mass spectra of the SV-OOA and LV-OOA derived here agreed well with the two background OOAs derived from PMF analysis of the whole dataset, including the clean periods discussed in this study and the periods influenced by wildfires (Zhou et al., 2017) (Figs. S4 and S5; $r^2$ > 0.9)."

Line 151: The authors write that NH4 and SO4 are relatively comparable in the FT and BL, but this seems to contradict the acidity gradient described later (based on NH4/SO4 molar ratios), please clarify. Furthermore, the conclusion that the sources of SO4 are the same in the FT and BL is not really supported by the different size distributions observed (and the explanation given there). Given that sulfate concentrations in the BL are going to be a strong function of BL height and FT exchange, a direct comparison of these concentrations is not very meaningful, suggest removing.

Thanks for point this out; we said it wrong. SO$_4$ mass concentration was comparable in the FT and BL (0.35 vs 0.33 ug/m$^3$) but NH$_4$ mass concentration in the BL was on average 1.6 times of that in the FT, therefore, yielding more acidic particles in the FT. In addition, the fact that SO$_4$ size distribution was different in FT than in BL-influenced air suggests different sources of SO$_4$ in the FT and BL. Related texts have been revised:
"In contrast, sulfate exhibited relatively constant concentrations (Fig. 1e) and a less pronounced diurnal pattern (Fig. 2). The weaker influence from BL evolution indicates similar sulfate concentrations in the BL and FT in the remote continental region of the western US. This is consistent with the relatively long atmospheric lifetime and the regional characteristics of sulfate particles."

Line 166/Fig 2: Constructing diurnal profiles based on six days of data is less than ideal, and the trends are somewhat obscured by the different times at which the PBL rose above the sampling site . Again, I suggest extending the PMF analysis for the full length of the deployment (Line 220) to spot check at least for some of the variables the robustness of the trends shown. That might also allow to filter for days where the BL/FT switch happened at consistent times and hence improve the robustness of the aerosol trends

We agree that more data would help the statistics. This study covered more than 8 days of data, with a time resolution of 2.5-min or 5-min. There are a total of 2192 data points included in the diurnal analysis.

The box whisker plots of the diurnal profiles of aerosol species and properties were examined. The results suggest that the diurnal trends are robust. Figure 2 shows the median diurnal trends instead of the mean, which further lowers the influences from episodic events. In addition, we performed a separate PMF analysis covering the full 1-month duration of the study. Periods with organic concentration below 1.5 µg m$^{-3}$, which hindered the model to converge, were excluded from PMF analysis (Zhou et al., 2017). Approximately ~ 50% of the data during the clean periods were included in the full length PMF analysis. Two background OOAs were derived from the full length PMF analysis. The temporal trends and mass spectra of the SV-OOA and LV-OOA derived from this study (clean periods only) agreed well with the full length PMF results (Fig. S4 and S5). The diurnal profiles of SV-OOA and LV-OOA also highly resembled those derived from the full length PMF analysis. These results further support that the diurnal profiles shown in Fig. 2 are representative of the background aerosols at MBO.

Line 186: It should be mentioned that Lee et al, "Substantial secondary organic aerosol formation in a coniferous forest: observations of both day- and nighttime chemistry", Atmos. Chem. Phys., 16, 6721–6733, 2016, doi:10.5194/acp-16-6721-2016, reported the same trends in particulate organic nitrated at the Whistler "middle altitude site" with likely very similar forests sources as at MBO . The authors may also consider adding this dataset to Fig 6 as well, although the lower sampling altitude makes it less comparable to other high altitude observations.
We thank the reviewer for this suggestion. We have added the following text to the end of the paragraph: "Similarly, Lee et al. (2016) observed that organonitrates made a significant contribution to the secondary OA (SOA) mass in the coniferous forested regions at Whistler – a mid-altitude site in western Canada."

With regard to the suggestion of including the dataset in Fig. 6, we chose not to considering that the dataset was obtained at a mid-mountain site at lower sampling altitude, thus is less comparable to the other mountaintop datasets shown in Fig. 6.

Line 196: The MSA analysis is well done and a nice addition to the manuscript. Are the authors aware of the study by Sorooshian et al, "Surface and airborne measurements of organosulfur and methanesulfonate over the western United States and coastal areas" J. Geophys. Res. Atmos., 120(16), 8535–8548, doi:10.1002/2015JD023822, 2015? They measured MSA and OS at ground sites near MBO during the Summer of 2013 as well, and found broadly similar concentrations, consistent with MSA being made in the continental BL.
We thank the reviewer for this reference. We have added the following sentence at the end of the 2$^{nd}$ paragraph in Section 3.2:
"Sorooshian et al. (2015) measured MSA and organosulfates at inland ground sites near MBO and found broadly similar concentrations."

We have also revised the last paragraph in Section 3.2., which now reads:
      "Oceans are generally considered a dominant source of dimethyl sulfide (DMS) and therefore its oxidation product MSA. However, the Pacific Ocean is 195 km to the west of MBO whereas the bivariate polar plot of MSA revealed that high concentrations were associated with winds from the east and the south – the inland areas (Fig. S11). In addition, MSA concentrations showed a clear diurnal cycle with a substantial daytime increase (Fig. 2), which suggests significant sources from the PBL. Aerosols in the PBL over this region likely have negligible oceanic influences since the Cascades mountain range lies between the Pacific Ocean and Mt. Bachelor and may obstruct surface wind bringing marine emissions inland. These results suggest that the sources of MSA at MBO were mostly continental, where a wide range of terrestrial sources including soil, vegetation, freshwater wetland, and paddy fields can emit DMS (Watts, 2000 and references therein). Furthermore, the maximum MSA/SO$_4$ ratio in this study was ~ 0.081, much

lower than those observed in marine aerosols (e.g., average = 0.23 in sub-Arctic North East Pacific Ocean (Phinney et al., 2006)). Similarly, lower MSA/SO$_4$ ratios were usually found in terrestrial regions, e.g., 0.01 - 0.17 in Fresno where MSA was mostly attributed to non-marine sources (Ge et al., 2012; Young et al., 2016), 0.007 - 0.15 along the Atlantic coast under continental influences (Zorn et al., 2008; Huang et al., 2017), and averages of 0.02 - 0.04 (maximum = 0.11) in California inland regions (Sorooshian et al., 2015)."

Section 3.4: The diurnal profiles show a fairly clear "transition zone" between FT and BL influence, and including/excluding those periods will likely strongly affect the averages found (see also Wagner et al, Atmos. Chem. Phys., 15, 7085–7102, 2015 as an example on the difficulty of properly defining the top of the BL). Such an analysis, either based on the diurnal profiles themselves or on the uncertainty of their threshold criteria, would help strengthen the confidence in the reported averages and trends, especially for the FT data.

The segregation of FT periods and BL-influenced periods in this study was performed based on water vapor mixing ratio. As discussed in Section 1 in the Supplement, extensive work has been done at MBO to differentiate FT air from BL influenced air using water vapor mixing ratio. Weiss-Penzias et al. (2006) and Fischer et al. (2010) used percentiles of water vapor mixing ratio to identify FT and BL-influences air masses. These same studies also used water vapor calculated from the 0 and 12 UTC National Weather Service soundings from Medford, Oregon and Salem, Oregon to determine a representative altitude for the air masses sampled at MBO. Reidmiller et al. (2010) used chairlift soundings of water vapor mixing ratios at Mt. Bachelor coupled with high-resolution NOx data from MBO to determine a time of day when the BL influence began at the mountain summit. While all these studies focus on segregating FT and BL-influenced air masses during Spring, Ambrose et al. (2011) used sounding data from Medford to compare with water vapor distributions at MBO and obtained a seasonal mean water vapor mixing ratios for MBO FT data, which for winter, spring, summer, and fall seasons corresponded with WV < 3.28, < 3.28, <5.4, and <4.12 g kg$^{-1}$. Based on the same work by Ambrose et al. (2011), McClure et al. (2016) interpolated between the seasonal values into a monthly criterion. Zhang and Jaffe (2017) contributed to a more accurate monthly FT/BL isolation at MBO based on comparison of MBO water vapor distributions to the water vapor soundings from Medford and Salem, Oregon, at equivalent pressure level. The water vapor criterion for FT air masses at MBO refers to the cut points when equivalent monthly averages of the retained drier portion of the MBO data and the soundings were obtained. The FT/BL-influenced water vapor criterion for each month is 3.26, 2.64, 2.46, 2.55, 3.06, 4.25, 5.14, 5.23, 4.60, 4.36, 3.44, 2.97 g kg$^{-1}$ (from January to December).

The water vapor cut points for FT periods in July and August were 5.14 and 5.23 kg$^{-1}$, respectively. However, Convection in summer enhances the vertical transport of air masses and creates a thicker entrainment zone. The top of the BL is therefore even more challenging to define. As FT air is drier than BL air, a lower water vapor threshold would reduce the influence of entrainment zone on "FT" isolation. Therefore, we applied a more stringent water vapor criterion, the driest (minimum) segregation value of 2.5 g kg$^{-1}$.

We have revised the first paragraph of Section 3.4 to clarify these points. It now reads:
    "To further examine the differences between aerosols in the free troposphere and boundary layer, we segregate periods using measurements of water vapor (H$_2$O$_{(g)}$). Extensive work has been done to differentiate free tropospheric air from boundary layer-influenced air at MBO using water vapor chairlift soundings (Reidmiller et al., 2010) and other approaches (Weiss-Penzias et al., 2006; Fischer et al., 2010; Ambrose et al., 2011; McClure et al., 2016; Zhang and Jaffe, 2017), as discussed in more detail in Section 1 of the Supplement.  Zhang and Jaffe (2017) established more accurate monthly H$_2$O$_{(g)}$ criteria for FT air masses at MBO – 5.1 and 5.2 g kg$^{-1}$ for July and August, respectively, and associated FT air masses with low H$_2$O$_{(g)}$ values. However, since convection in summer enhances vertical transport and creates a thicker

entrainment zone where BL mixed with FT, properly defining the top of BL is challenging (Wagner et al., 2015). To avoid the influences of the transition zone on FT, we used a more stringent $H_2O_{(g)}$ criterion, 2.5 g $kg^{-1}$, which is the lowest monthly cut point reported in Zhang and Jaffe (2017). In addition, we explored the usage of the estimated BL height from HYSPLIT back trajectory analysis as the segregation criteria. A comparison between these two methods is available in Section 1 of the Supplement. After careful evaluation, we classify periods with $H_2O_{(g)}$ < 2.5 g $kg^{-1}$ and CO < 80 ppb as "FT air" and the rest as "BL-influenced air".".

Line 275-278: This seems mostly speculative. MSA could also be oxidized further to sulfate and/or partition to the gas phase instead of being washed out. Consider removing/shortening.
We have revised and shortened the discussions. The paragraph now reads:
"MSA correlated with HR-AMS sulfate for different aerosol regimes with different slopes. As shown in Fig. 4b, BL-influenced aerosols showed a range of $MSA/SO_4$ ratios generally higher than FT aerosols. This may be attributed to higher MSA concentration near terrestrial sources in the BL. Indeed, airborne measurements of MSA in aerosol over the western US in summer 2013 have shown that MSA loading decreased with the increase of altitude (Sorooshian et al., 2015). Furthermore, as discussed later on, sulfate was likely produced in the FT during regional new particle formation and growth events, which may further contribute to lower $MSA/SO_4$ ratio in FT aerosols."

Line 295-306: Again this seems fairly speculative. There is no discernible enhancement of small particles in the (fairly noisy) sulfate size distribution presented, which is still within the envelope of the BL size distribution. So preferential activation/wash out of larger particles would suffice to account for the observed differences and nucleation is not needed to explain the difference. So consider shortening/removing.
We have revised the discussion and the texts now read:
"A distinctly different size distribution was observed for sulfate-containing particles in the FT (Fig. 5c), which exhibited a prominent mode at ~ 250 nm. One possible explanation is preferential activation/wash out of larger particles, and thus the reduction of sulfate signal in larger size (droplet) modes. Condensational growth of newly nucleated particles in the FT may be another possibility. Scavenging could result in low particle surface area, which facilitates new particle formation (NPF) in the FT. Although we did not observe NPF events in this study (due to instrumental limitations), in-situ NPF events have been frequently observed in the FT (e.g., Hallar et al., 2011; Hallar et al., 2013). Formation and growth of new particles have also been observed over broad regions in the FT (Tröstl et al., 2016). Condensation of gas-phase sulfate products on small FT particles could contribute to the observed condensation mode sulfate particles at MBO. These observations may shed light on the different sources and processes of aerosols in the BL and FT and suggest that sulfate and organic aerosols were likely present in both internal and external mixtures at MBO."

Line 328-336: I am not following. If the average FT NH4/SO4 molar ratio is less than 0.3, the particles will be liquid bisulfate/sulfuric acid down to fairly low RH, hence the discussion of solid ammonium sulfate as presented does not seem really relevant to the findings. Please explain.
We have revised the discussion, which now reads:
"The extent to which sulfate particles are neutralized has major implications for aerosol radiative forcing. The average relative humidity at MBO was 25.6 (±8.9) % during the clean periods. Acidic sulfate aerosols are more hygroscopic than ammonium sulfate. The resulting increase in aerosol water content both increases the direct radiative forcing of sulfate (Adams et al., 2001; Jacobson, 2001) and promotes homogenous ice nucleation (Koop et al., 2000). In addition, while mineral dust particles coated with

ammonium sulfate are efficient ice nuclei, those coated with sulfuric acid can lose their ice nucleating ability (Eastwood et al., 2009). **"**

Minor comments:
Line 32: "At lower altitudes " instead of "in lower altitudes"
Changed.

Line 39: While aircraft measurements are indeed expensive, I would argue that the main advantage of mountaintop observatories are long-term, continuous measurements that are invaluable for statistics. Obviously deploying complicated instrumentation such as the AMS can be a challenge for such sites as well, but I think it is worth mentioning this.
Agree. The following sentence has been added:
"Another main advantage of mountaintop observatories are long-term continuous measurements, which are invaluable for statistics."

Line 40: Given the inclusion of the Whistler site, I would suggest replacing "US" with "North America"
Replaced.

Line 45: Suggest adding to your list of non-AMS particle measurements: L. Ahlm et al, "Temperature-dependent accumulation mode particle and cloud nuclei concentrations from biogenic sources during WACS 2010", Atmos. Chem. Phys., 13, 3393–3407, 2013, doi:10.5194/acp-13-3393-2013
Added.

Line 82: "the established data analysis tool" instead of "established data analysis tool"
Changed.

Line 84: It would be useful to document the range of CEs observed. A histogram keyed by BL/FT influence would be appropriate, and would serve to highlight again the gradient in acidity between the different air masses sampled at MBO.
The composition-dependent collection efficiency ranged from 0.5 to 1 during this study. A histogram of the CE values colored by FT and BL-influenced periods is shown below. High CE was more frequently observed during FT periods, mainly because of acidic particles.

[Figure]

We have revised the text to read:
"A composition-dependent CE (ranging from 0.5 to 1; average = 0.66) was applied based on…"

Line 131: Is the chloride associated with periods of somewhat larger inorganic nitrate?
As Hu et al, Aerosol Sci. Technol., 51,735-754, doi:10.1080/02786826.2017.1296104 described, under such conditions chloride can be a vaporizer artifact, and is easily testable by reviewing the ammonium nitrate calibrations. Given the relative concentrations of chloride and nitrate, this is unlikely, but should be mentioned.

We didn't see elevated chloride signal associated with high nitrate mass. In fact, for most of the time during the clean periods, chloride signal was below limit of quantification (i.e., < 3 times of detection limit). We therefore removed the mentioning of chloride in aerosol composition. We have added the following sentence in the revised manuscript: "Chloride was close or below detection limit for most of the time during the clean periods."

Fig 4: Suggest adding a timeseries of the ratios, with the FT periods clearly marked, that would make the point(s) better than these correlation plots.
The time series of the ratios are shown below. We think the points we want to convey are more clearly illustrated in the scatterplots though.

[Figure]

Fig S13 (middle): Please add the background line from Cubison et al. It looks like your factors lie pretty much right on top of it, as in Zhou et al, 2017.
Done as suggested. Yes, both factors situate to the left of the background line ($f_{60}$ = 0.3%).

**References**

Ambrose, J. L., Reidmiller, D. R., and Jaffe, D. A.: Causes of high O3 in the lower free troposphere over the Pacific Northwest as observed at the Mt. Bachelor Observatory, Atmos. Environ., 45, 5302-5315, 2011.

Fischer, E. V., Jaffe, D. A., Reidmiller, D. R., and Jaegle, L.: Meteorological controls on observed peroxyacetyl nitrate at Mount Bachelor during the spring of 2008, J. Geophys. Res. Atmos.,, 115, 2010.

Lee, A. K. Y., Abbatt, J. P. D., Leaitch, W. R., Li, S. M., Sjostedt, S. J., Wentzell, J. J. B., Liggio, J., and Macdonald, A. M.: Substantial secondary organic aerosol formation in a coniferous forest: observations of both day- and nighttime chemistry, Atmos. Chem. Phys., 16, 6721-6733, 2016.

McClure, C. D., Jaffe, D. A., and Gao, H.: Carbon Dioxide in the Free Troposphere and Boundary Layer at the Mt. Bachelor Observatory, Aerosol Air Qual. Res., 16, 717-728, 2016.

Reidmiller, D. R., Jaffe, D. A., Fischer, E. V., and Finley, B.: Nitrogen oxides in the boundary layer and free troposphere at the Mt. Bachelor Observatory, Atmos. Chem. Phys., 10, 6043-6062, 2010.

Sorooshian, A., Crosbie, E., Maudlin, L. C., Youn, J.-S., Wang, Z., Shingler, T., Ortega, A. M., Hersey, S., and Woods, R. K.: Surface and airborne measurements of organosulfur and methanesulfonate over the western United States and coastal areas, Journal of Geophysical Research: Atmospheres, 120, 8535-8548, 2015.

Weiss-Penzias, P., Jaffe, D. A., Swartzendruber, P., Dennison, J. B., Chand, D., Hafner, W., and Prestbo, E.: Observations of Asian air pollution in the free troposphere at Mount Bachelor Observatory during the spring of 2004, J. Geophys. Res. Atmos.,, 111, D10304, 2006.

Zhang, L. and Jaffe, D. A.: Trends and sources of ozone and sub-micron aerosols at the Mt. Bachelor Observatory (MBO) during 2004–2015, Atmos. Environ., 165, 143-154, 2017.